# How Relevant Is the Sentence Unit to Accessing Implicit Meaning?

Céline Pozniak [1], Claire Beyssade [1], Laurent Roussarie [1] and Béatrice Godart-Wendling [2,3,*]

1   Department of Linguistics, University of Paris 8, 93200 Saint-Denis, France;
    celine.pozniak@univ-paris8.fr (C.P.); claire.beyssade@gmail.com (C.B.); laurent.roussarie@univ-paris8.fr (L.R.)
2   The Institute of Legal and Philosophical Sciences, Paris 1 Panthéon-Sorbonne University, 75005 Paris, France
3   The French National Center for Scientific Research, 75016 Paris, France
*   Correspondence: beatrice.godart-wendling@cnrs.fr

**Abstract:** This paper examines the relevance of the sentence concept to the understanding of three types of implicitness (presupposition, conversational implicatures, irony). Our experimental protocol involved 105 children (aged 6 to 11) and 82 adults who were asked to read short texts composed of a context about some characters and a target sentence conveying one of the three implicit contents. After reading, children and adults had to answer a comprehension yes-no question and indicate the segments from the text that helped them answer the question. Results showed a difference between the three types of implicitness, with presupposition being detected and understood at a subsentential level, whereas implicatures and irony come under extrasentential level requiring the context to be taken into account. Referring to sentence as a unit of meaning does not seem relevant as soon as understanding is not limited to the literal meaning of what is written, but also concerns what is meant by the text.

**Keywords:** implicitness; sentence; presupposition; conversational implicature; irony

## 1. Introduction

It is well known that many syntactically full-fledged sentences in a text can be perceived as semantically incomplete in so far as they contain underspecified meanings, context-sensitive expressions, or implicit contents.[1] Think of pronouns or quantified expressions, which cannot be interpreted without the addressee retrieving from the context the antecedent of the pronoun or the quantification domain. In this paper, we are specifically interested in the implicit meaning conveyed by a text, which has been shown to play a central role in comprehension. Indeed, understanding a text is not just a matter of understanding the literal meaning of each sentence that makes it up, but of gaining access to the elements of implicit meaning that make it possible to turn this sequence of sentences into a cohesive whole (by reconstructing the identity and inclusion relationships between the referents mentioned) and a coherent whole (by reconstructing the temporal and logical links between the situations described). So even if the sentence constitutes, one way or another, a unit of thought, the question we aim to address here is: Does it constitute a "unit of understanding" for the pupils? More precisely, how relevant is the concept of a sentence when it comes to retrieving the implicit information that contributes to the global coherence of texts the pupils have to read?

By implicitness, we mean the covert pieces of information that are necessary to reach an accurate understanding of what is meant in a text. We will call "implicit" the elements of meaning that are inferred (in a broad sense) from what is said, without having been explicitly said. Implicit contents are of a very varied nature. Since Ducrot (1972), it is usual to distinguish the linguistic or literal implicit, associated with presuppositions, from the discursive implicit, corresponding to what Grice ([1967] 1975) referred to as implicatures.

Presuppositions are characterized by the presence of a trigger, a linguistic element (a word or a grammatical construction) which is part of what is literally said. It is because the speaker uses the verb *continuer* 'continue' in (1a) that the addressee makes the inference (1b). As one can continue only what was started before, *continuer* 'continue' presupposes having started. The source of the presupposition is lexical: the presence of the word in the statement is enough for the presupposition to emerge.

(1)   a.   Tom continue à faire des progrès en calcul.
            'Tom continues to make progress in arithmetic.'
      b.   Tom avait déjà fait des progrès en calcul.
            'Tom had already made some progress in arithmetic.'

Implicatures also convey implicit information. They are non logical inferences: "A implicature is a proposition that is implied by the utterance of a sentence in a context even though that proposition is not a part of nor an entailment of what was actually said" (Gazdar 1979, p. 38). In this respect, implicatures differ from logical consequences. Grice ([1967] 1975), who introduced the term *implicature*, distinguishes conventional implicatures (about which he says little and which we don't consider here) from conversational implicatures. He shows that the content of a conversational implicature is not attached to a particular linguistic form (and in this respect implicatures differ from presuppositions) but is the result of the articulation of a content and a general principle of interpretation: the Cooperative Principle. Knowledge of this principle, mutually shared by the dialogue participants, leads them to make assumptions about Speaker intentions and enables them to derive supplementary meaning associated with an utterance. Grice considers the Cooperative Principle as the foundation of the rules that govern conversation in its ordinary use, what he calls Maxims of Conversation. Example (2) illustrates how participants in the conversation rely on the Principle of Cooperation and the Quantity Maxim to derive the implicature (2b). The addressee must engage in a reasoning process which is based both on the use of a lexical element (here the numeral *two*) and on a conversational rule derived from the Quantity Maxim, which stipulate in substance that the speaker has said everything he could say, no more, no less. If (2a) generates the implicature (2b), it is because the speaker used the numeral *two* rather than the immediately following numeral, *three*, and that numerals form a scale. (2b) exemplifies a case of scalar implicature[2]. However, the source of the implicit in (2a) cannot be reduced to a lexical element; mastery of conversational uses and the implementation of reasoning based on the comparison of the entire statement with other possible statements are also required. Furthermore, it has to be noted that implicatures are defeasible inferences, which means that they are only probable; they are drawn in situations, as long there is no indication to the contrary

(2)   a.   Tom a réussi deux exercices.
            'Tom successfully completed two exercises.'
      b.   Tom a réussi deux exercices et pas plus.
            'Tom successfully completed two exercises and no more.'
      c.   Tom a réussi deux exercices, même trois si je me souviens bien.
            'Tom successfully completed two exercises, even three if I remember correctly.'

In addition to these two types of implicit meaning, there is a third type, which differs from the previous ones in that it is not associated with the use of a particular lexical item. This third type of implicitness is typically attached to stylistic devices, such as irony, which can lead to giving a statement an interpretation diametrically opposed to its literal meaning. This is the case with (4a) which, in context (3), loses its literal and compositional meaning and means (4b), the exact opposite, instead. Generally speaking,[3] irony is viewed as the act of meaning something by saying something quite different, while making one's communicative intention clear enough to be recovered. In ironic statements, there is no enrichment of the literal meaning (contrary to usual implicatures), but rather the substitution of a

completely different meaning in place of the literal meaning. And such a substitution is obligatory: The addressee needs to achieve it lest the whole text appear globally incoherent. To do this, they must take into account the context. Irony is by nature context-sensitive; the same sentence, depending on the context, will or will not be interpreted as ironic. To interpret a statement as ironic, the addressee has to look outside the limits of the ironic sentence.

(3)    Tom n'aime pas aller à l'école. Il a beaucoup de mauvaises notes.
       Ce week-end, ses parents doivent signer ses cahiers. Son père lui dit:
       'Tom doesn't like going to school. His grades are poor. This weekend,
       his parents have to sign his exercise books. His father said to him:'

(4)    a.  Alors, toujours le meilleur de la classe?
           'So, still top of the class?'
       b.  Tu nous rapportes encore de mauvaises notes?
           'Have you brought bad grades home yet again?'

In this paper, we focus on these three types of implicitness (presuppositions, generalized conversational implicatures, and irony) because the psycholinguistic literature has shown that they are already cognitively accessible to children from the age of five in favorable contexts[4] (Scoville and Gordon 1980; Pouscoulous et al. 2007; Loukusa and Leinonen 2008; Eiteljoerge et al. 2018). We are interested in what, in the process of interpretation, really triggers the understanding of these three types of implicitness both among primary school students and among adults. We chose to approach this issue by asking ourselves to what extent the concept of sentence/sentential unit is relevant and privileged in this comprehension task. In particular, are the elements perceived as responsible for implicit meanings (the triggers) located within the sentence which conveys implicit contents? Or, on the contrary, is the processing of these implicit meanings carried out at another level than the sentence, relying on the understanding of segments that are either smaller (such as words or phrases) or larger (such as the context)?

As a starting point, we considered the following three theoretical hypotheses.

(1) Presuppositions are usually triggered by lexical items or specific syntactic constructions (such as clefts for example), so they can be detected on a very local scale and at a subsentential level. In this case, we expect a more local comprehension, with phrases being more relevant than sentences for understanding presuppositions.

(2) Conversational implicatures can also be associated with particular linguistic material, but they are defeasible inferences that are computed by reasoning on the maxims of conversation, the speaker's presumed intentions, and the relevant surrounding information; they are more context-sensitive and should be detected on a less local scale, within larger text spans possibly of sentential dimension (in particular sentences often constitute a minimal relevant domain to check whether or not an implicature is canceled or suspended). Compared to presuppositions, we expect a less local comprehension, and segments larger than phrases should be necessary for understanding implicatures.

(3) As for irony, its understanding requires that the addressee/reader recognizes a form of discordance, dissociation, or pretense by the speaker between what is said and what is actually meant (Garmendia 2018). In this case, the addressee needs to be much better acquainted with the context of the utterance to be able to make such speculations, and one can assume that irony will rarely be detected at the level of the single statement or sentence bearing it; a larger extent of context must be taken into account, especially in written texts. Concretely, we expect a more global comprehension of irony, with large parts of the text being important for understanding this type of implicitness.

To test these hypotheses, we designed an experiment whose goal was twofold: (i) to observe how these three kinds of implicit contents are correctly understood by pupils throughout their schooling and by adults and (ii) to identify the linguistic clues that enable them to draw these inferences. More specifically, we looked at where the words or chunks of words that participants considered relevant to justify their understanding of the implicit contents occurred in the text: within the target sentence (containing the implicit information)

or in the preceding context. This enabled us to address the questions of whether the concept of sentence is intuitively perceived as a necessary "unit of understanding" with regard to implicitness and whether it is perceived as a sufficient one.

If, in order to justify their answers, participants (either children or adults) select more often segments from the context, we can infer that understanding the sentence carrying the implicit is not, in itself, sufficient to construe the implicitness. If participants select segments from the context less often, we can assume that understanding the sentence suffices to detect the implicit meaning.

With regard to the selection made by participants in the target sentence, if segments from the target sentence are less often selected, and if these are precisely the triggers of presuppositions or implicatures, then we can conjecture that a global comprehension of the sentence is not absolutely necessary for recognition of the implicit meaning. More precisely, this would indicate that correct understanding by children of certain implicit contents depends above all on their semantic mastery of specific lexical items or syntactic construes, which have been identified locally. Conversely, if segments are more often selected within the target sentence, we can infer that a global comprehension of the sentence is a necessary step for recognizing the implicit meaning and that detecting local triggers alone is not sufficient. Note that to diagnose the necessary nature of sentence comprehension, we did not expect participants to select every segment in the sentence. We left the participants free to consider that certain segments might be of secondary importance in justifying their responses. Our initial predictions, in line with the theoretical hypotheses mentioned above, were the following:

For the presuppositions, we expected that global comprehension of the target sentence would appear to be sufficient but not necessary. We expected that the sentence as a unit would not be perceived as relevant and that only segments from the target sentence would be more often selected.

For irony, we expected that global comprehension of the target sentence would be necessary but not sufficient and that the context would be heavily called upon (i.e., segments from the context would be more often selected).

For implicatures, our prediction was more complex: we expected global comprehension of the target sentence to appear more often as not sufficient (to emphasize the role of context) but also less often as necessary (to emphasize the locality of the implicature). For example, we would expect a more often selection of segments in the context sentence than for presupposition, but still less often than for irony.

The paper is organized as follows. In Section 2, we describe the experimental method we used to answer the question we address: what role does the sentence play as a unit of understanding in the detection of implicitness? To do this, we provide details about the participants, the material and the experimental design, and the analysis method. Then, we present in Section 3 the results concerning on the one hand the understanding of the different types of implicitness and on the other hand the parts of the text that the participants identify as having allowed them to interpret implicit meaning. In Section 4, we discuss the results before concluding the paper.

## 2. Method

### 2.1. Participants

The experiments were conducted on a university-hosted instance of Ibexfarm (Internet-Based Experiments Farm, see Drummond 2013). For the adults, 84 participants (mean: 34 y.o, $\sigma = 11$) were recruited on the Prolific platform (www.prolific.co) where they were compensated around 3 pounds for 20 min. Two participants were excluded because they were not early monolingual speakers of French. For the children, 137 participants were recruited in a state primary school in Paris (13th arrondissement).[5] Because of two technical problems (children not writing down their age and lists not well counterbalanced across the different classes[6]), we had to exclude 32 participants, leaving us with 105 participants in different classes (see Table 1).

**Table 1.** Distribution of the children in the different classes.

| Grade 1: CP (Around 6 y.o.) | Grade 2: CE1 (Around 7 y.o.) | Grade 3: CE2 (Around 8 y.o.) | Grade 4: CM1 (Around 9 y.o.) | Grade 5: CM2 (Around 10 y.o.) |
|---|---|---|---|---|
| 21 | 15 | 24 | 24 | 21 |

*2.2. Materials and Design*

We created 12 short texts that were composed of a context about some characters and a target sentence conveying implicit content (see Table 2 for an example). The manipulated variable was the type of implicitness: presupposition, conversational implicature, and irony, leading to three conditions per item (following a Latin-Square design, with three versions—lists—of the experiment).

**Table 2.** Example of an experimental item under the three conditions.

| Conditions | Context | Target Sentence | Question |
|---|---|---|---|
| Presupposition | Tom n'aime pas aller à l'école. Il a beaucoup de mauvaises notes. Ce week-end, ses parents doivent signer ses cahiers. 'Tom doesn't like going to school. He gets a lot of bad grades. This weekend, his parents have to sign his exercise books.' | Ils sont contents parce que Tom continue à faire des progrès en calcul. 'They are pleased because Tom continues to make progress in arithmetic.' | Est-ce que c'est la première fois que Tom fait des progrès en mathématiques ? 'Is this the first time Tom has made progress in math?' |
| Implicature | | Ils sont contents parce qu'il a réussi deux exercices. 'They are pleased because he managed to do two exercises.' | A votre avis, est-ce que Tom a raté tous les autres exercices ? 'In your opinion, did Tom fail to do the other exercises?' |
| Irony | | Son père lui dit: "Alors, toujours le meilleur de la classe ?" 'His father said to him: "So, still top of the class?"' | À votre avis, est-ce que le père de Tom pense que son fils est le meilleur de la classe? 'In your opinion, does Tom's father think his son is top of the class?' |

For presuppositions and implicatures, each target sentence contained a specific item (henceforth the expected word segment) that corresponds to the presupposition trigger or the lexical locus of the implicature, respectively (for instance *continue* 'continue' and *deux* 'two' in Table 2 above). The presuppositions in the target sentences were intentionally designed to be globally unbound (in the sense of van der Sandt 1992) in the whole text. It means that the presuppositional contents were never related to any clear and explicit antecedent in the context; thus, the correct understanding of the presuppositions necessarily went through a global accommodation. This was to ensure that the presupposition triggers were properly recognized as such and avoid any interference from the contexts (if the presuppositions had been bound to an antecedent in the context, then their content would have appeared overtly in the text, and they could not have counted as genuine implicit meanings). In this way, we took presuppositions (and in particular, *semantic* presuppositions) as a device to test a specific kind of implicitness, namely the implicit meanings that are closely related to linguistic clues and do not necessitate an important appeal to conversational principles to be recovered. On the other hand, implicatures and irony were handled to test a more pragmatic kind of implicitness, involving more reasoning and more distancing from the literal meaning. The main difference between implicatures and irony here is that the failure to retrieve implicated content does not bring about a full misunderstanding of the utterance, whereas the failure to recognize an ironic statement leads to global incoherence.

After each text, there was a comprehension yes-no question to check whether the participant had inferred the implicit content from the text (Table 2). The number of correct 'yes' and 'no' answers was counterbalanced at the experiment level (not at the list level). For the children, the text remained on the screen during question answering to make the task easier for them. For the adults, the display of the text was manipulated between participants: half the participants continued to see the text during question answering (as

in the experiment with the children), while for the other half, the text disappeared during question answering.

Sixteen fillers were added for the experiment with adults and consisted of small texts with the same form as experimental items, except that the type of implicitness was not manipulated (see Table 3 for an example). For both adults and children, there were two practice items at the beginning of the experiment that were similar to the fillers.

**Table 3.** Example of a filler.

| Content | Target Sentence | Question |
|---|---|---|
| Émilie et ses trois enfants vont au marché durant le mois de septembre. Elle essaie de faire attention à acheter des fruits et des légumes de saison. 'Émilie and her three children go to the market during the month of September. She tries to buy seasonal fruits and vegetables.' | Un de ses enfants se met à pleurer parce qu'il voudrait manger des fraises. 'One of her children starts crying because he wants to eat strawberries.' | A votre avis, est-ce qu'Émilie va au marché? 'In your opinion, does Émilie go to the market?' |

### 2.3. Procedure

Both the children and the adults were asked to carefully read the short texts and answer questions about them. Then, they were asked to drag and drop segments from the text that helped them answer the questions (Figure 1). The experiment lasted around 20 min for the adults and between 20 and 30 min for the children.

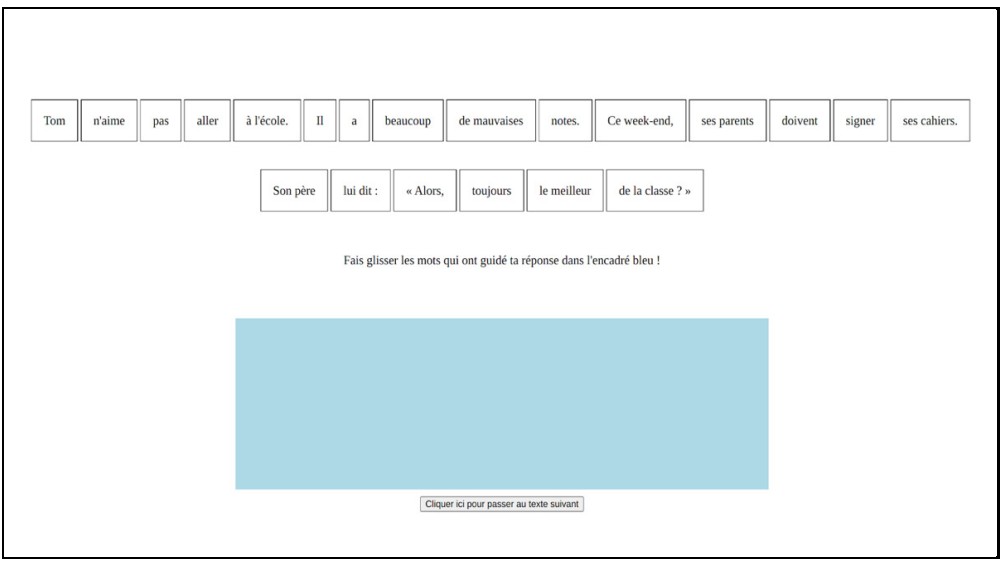

**Figure 1.** Example of how the word segments were presented in the experiment.

### 2.4. Method of Analysis

We ran Bayesian modeling with the R software (4.3.1 version, R Core Team 2020) and the Rstudio interface (Posit team 2023) using the Bayesian Regression Models and the Stan package (Bürkner 2017; Carpenter et al. 2017; Bürkner and Charpentier 2020).

We chose Bayesian analyses because of the multiple advantages they present for our data. Indeed, it is possible to fit a maximal random effects structure (Barr et al. 2013) without convergence failure, even with small datasets. Also, they directly test the likelihood of the hypothesis of interest, allowing us to go beyond the binary decision threshold (for more information about Bayesian analyses, see Kruschke 2014 or McElreath 2020).

For the answers to the questions, we analyzed our data using Bayesian binomial regression models with the Bernoulli family. The dependent variable was the answers

to comprehension questions (0 for incorrect answers and 1 for correct answers). For the selected segments, the dependent variable was the segments selected in the context, which remained the same through the three conditions (see Table 2). Moreover, we only looked at the selected segments of participants who had correctly answered the comprehension questions and when their mean accuracy was above 50%. We found that we had a large number of zero values (which means that no segment was selected). In order to have a simple statistical solution to allow correct interpretation of the data, we coded the dependent variable binomially (1 for the selected segment, regardless of precise number of selected segments, and 0 for no selection at all). We then analyzed our data using Bayesian binomial regressions with the Bernoulli family. For both children and adults, the independent variable was the type of implicitness (three levels: presupposition, implicature, and irony) with presupposition as the reference level. For the children, we added the class as another independent variable (transformed into a continuous variable). For the adults, we coded the display of the text (a between-participant variable) as 1 when the text was displayed during question answering and 0 when the text was not displayed. We applied mean-centered coding for each independent variable. Random variables were 'Participants' and 'Items'. For random variables, whenever relevant, we included the relevant random slopes as well as their interactions.

All Bayesian models generate a posterior distribution for the predictors. Here, we report the estimated mean (Est.), the range (95% credible intervals, that is to say, the probability that includes the true value of the predictor), and the probability of the effect of the predictor being smaller than (for negative estimates) or greater than (for positive estimates) zero ($P(Est. > 0)$ or $P(Est. < 0)$), a probability that there is an effect. We report results for which P(Est.) to be different from zero is >0.80.

## 3. Results

In this section, we present both the descriptive results (i.e., the proportions in the figures or the means in the text) and the inferential results from the statistical models (the estimated mean 'Est.', the 95% credible intervals 'CrI', and the probability of the effect 'P(Est. < or > 0)').

### 3.1. Comprehension Questions

Figures 2 and 3, respectively, show the proportions of correct answers for the children and adults. The color bars correspond to the three conditions: green corresponds to the texts with the presupposition condition, orange corresponds to the text with the implicature condition, and violet corresponds to the texts with the irony condition. The panels in pink in Figure 2 correspond to each class and the panels in pink in Figure 3 correspond to whether the text remained on screen during question answering (left panel) or if there was only the question (right panel). For each figure, 1 means that participants correctly answered the question, and 0 means that their answer to the question was wrong. For example, in Figure 2, children from the first group (mean age: 6.6 y.o.) had a mean accuracy of 0.32 in the irony condition, meaning that they had difficulty correctly answering the question compared to the presupposition condition, where their accuracy is 0.74 (i.e., closer to 1).

As can be seen in Figure 2 and confirmed in the analyses, children answered more correctly when they were in higher classes (the proportions increased in higher classes; inferential results: Est. = 0.28, CrI = [0.14, 0.42], $P(Est. > 0)$ = 1). As for the type of implicitness, we found that, compared to presupposition, questions about irony were less well understood (mean accuracy across all class levels: 0.58 for irony vs. 0.75 for presupposition), as were questions about implicature (overall mean accuracy: 0.61 for implicature vs. 0.75 for presupposition). These differences were confirmed by the statistical model (presupposition-irony comparison: Est. = −0.85, CrI = [−1.41, −0.31], $P(Est. < 0)$ = 1; presupposition-implicature comparison: Est. = −0.73, CrI = [−1.56, 0.09], $P(Est. < 0)$ = 0.96). However, when compared to presupposition, accuracy was better when children were in higher classes for questions about irony (the discrepancy between the proportions of irony

and presupposition are smaller in higher classes than in lower classes; inferential results: Est. = 0.33, CrI = [−0.02, 0.70], P(Est. > 0) = 0.97). This was not the case for questions about implicature compared to presupposition.

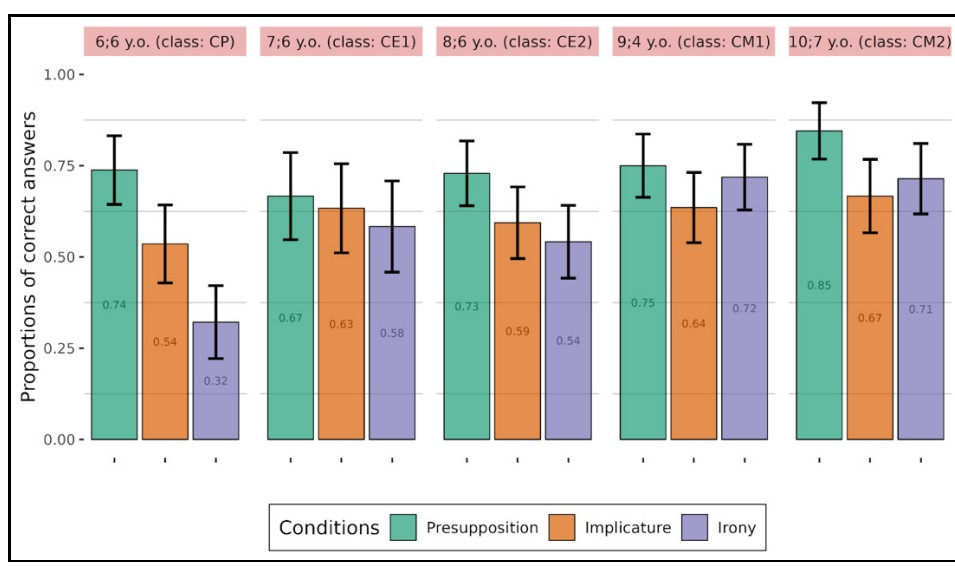

**Figure 2.** Proportions of correct answers depending on the type of implicitness and the class (children).

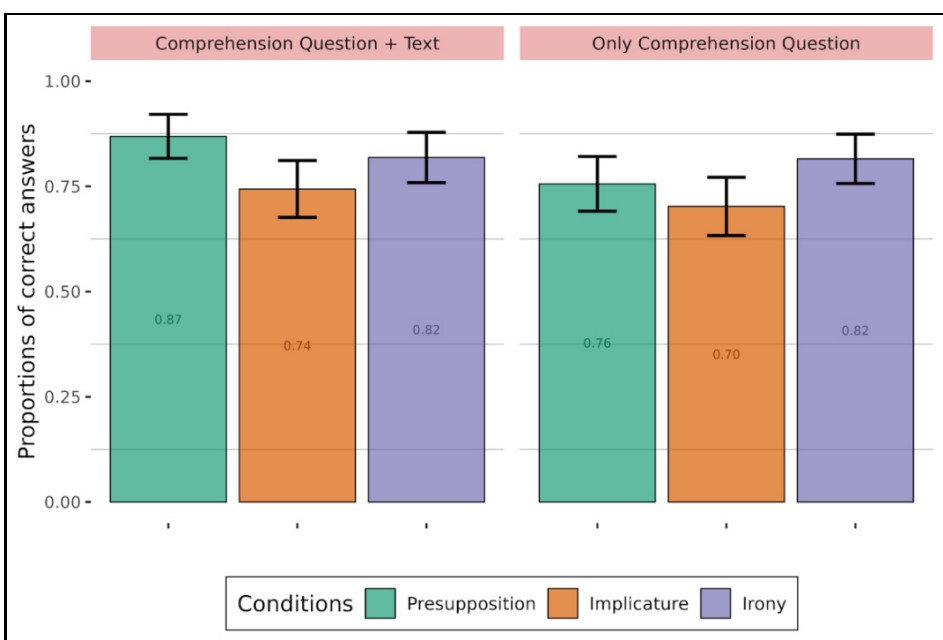

**Figure 3.** Proportions of correct answers depending on the type of implicitness and the presence or absence of the display of the text during question answering (adults).

As for the adults (Figure 3), we found that keeping the text on the screen during question answering improved accuracy in general (the overall proportions are higher in the left panel than in the right panel; inferential results: Est. = 0.63, CrI = [−0.09, 1.37], P(Est. > 0) = 0.96). Compared to presupposition, questions about implicature were less well understood (overall mean accuracy: 0.72 for implicature vs. 0.81 for presupposition; inferential results: Est. = −1.31, CrI = [−3.95, 1.17], P(Est. < 0) = 0.86) while the difference with questions about irony (mean accuracy: 0.82 for irony and 0.81 for presupposition) was negligible. We found an interaction between the presupposition-implicature comparison

and the display of text (inferential results: Est. = −1.35, CrI = [−2.94, 0.08], P(Est. < 0) = 0.97): questions about implicature were less well understood compared to questions about presupposition when the text was displayed during question answering (descriptively, in Figure 3, there was a difference of 0.06 between implicature and presupposition when the text was not displayed compared to a difference of 0.13 when the text was displayed). There was also another interaction between the presupposition-irony comparison and the display of text (inferential results: Est. = −1.62, CrI = [−3.34, −0.01], P(Est. < 0) = 0.98): answers from participants were less accurate for questions about irony compared to presupposition when the text was displayed, but it was the other way around when the text was not displayed, with greater accuracy for questions about irony compared to questions about presuppositions (see the pattern in Figure 3).

*3.2. Selection of Word Segments—Contexts*

For all analyses regarding the selection of word segments (either in the context or in the target sentence), we only looked at the selection when participants had correctly answered the comprehension questions and when their mean accuracy was higher than 50%[7]. For all results regarding the selection of word segments, it is important to keep in mind that 1 means that participants selected at least one word segment, while 0 means that participants did not select any word segment (whether in the context or in the target sentence).

Figures 4 and 5, respectively, show the proportions of word segment selection in the context depending on the conditions for children and adults.

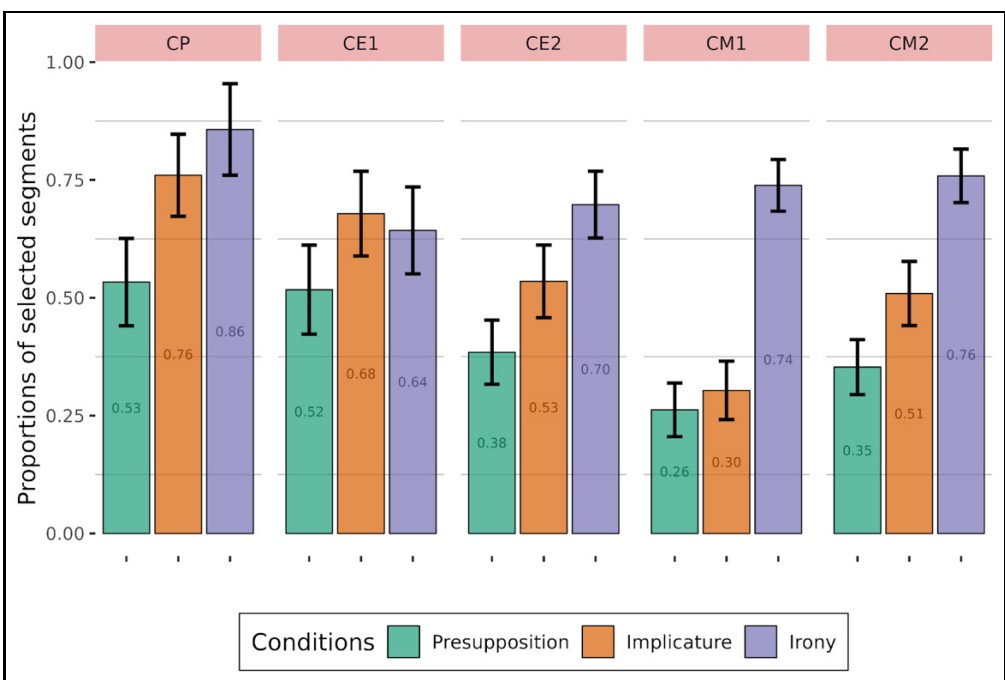

**Figure 4.** Proportions of segment selection depending on the type of implicitness and the class (children).

As for the children, in general, they selected segments in the context less often when they were in higher classes compared to lower classes (the proportions of selection are lower in higher classes—the right panels—in Figure 4; Est. = −0.34, CrI = [−0.97, 0.29], P(Est. < 0) = 0.86). As shown in Figure 3, in general, compared to presupposition, children selected segments more often for irony (mean selection across class levels: 0.73 for irony and 0.38 for presupposition; inferential results: Est. = 2.64, CrI = [1.25, 4.14], P(Est. > 0) = 1) and for implicature (mean selection across class levels: 0.51 for implicature and 0.38 for presupposition; inferential results: Est. = 0.77, CrI = [−0.83, 2.41], P(Est. > 0) = 0.84). When

looking at classes, as shown in Figure 4, compared to presupposition, children selected segments more often for irony when they were in higher classes (Est. = 0.43, CrI = [−0.42, 1.29], P(Est. > 0) = 0.85) but less often for implicature compared to presupposition (Est. = −0.37, CrI = [−1.06, 0.24], P(Est. < 0) = 0.89). As shown in Figure 4, this is due to the fact that children selected segments less often for presupposition when they were in higher classes.

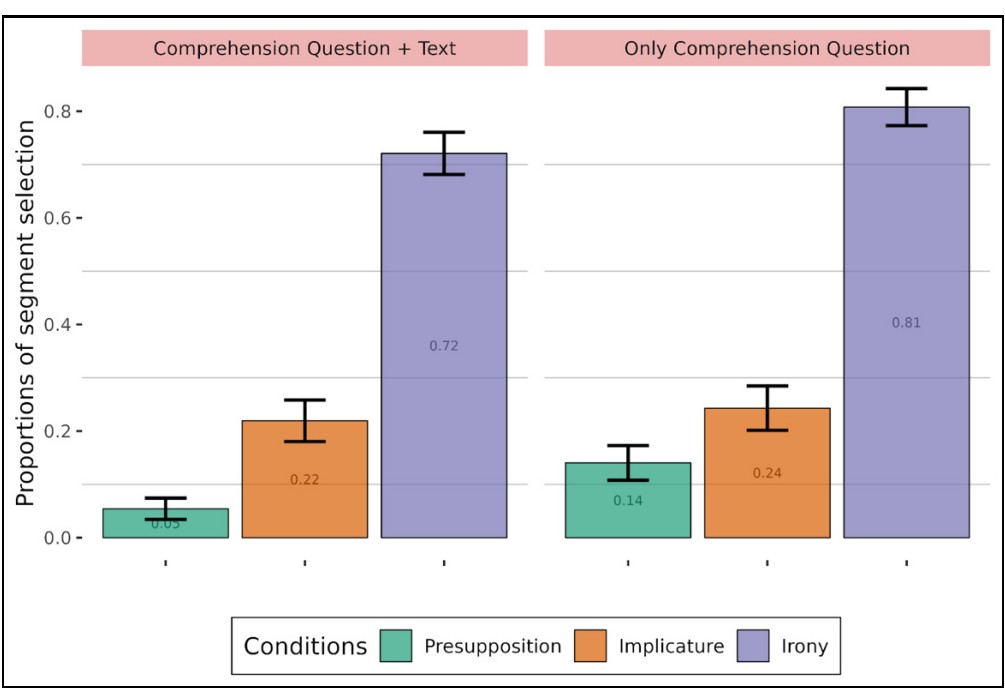

**Figure 5.** Proportions of segment selection depending on the type of implicitness and the presence or absence of the display of the text (adults).

As illustrated in Figure 5 and confirmed in the analyses, in general, adults selected segments in the context more often for irony compared to presupposition (overall mean selection: 0.76 for irony and 0.10 for presupposition; inferential results: Est. = 6.39, CrI = [3.87, 9.41], P(Est. > 0) = 1) and for implicature compared to presupposition (overall mean selection: 0.23 for implicature and 0.10 for presupposition, Est. = 1.21, CrI = [−1.55, 3.82], P(Est. > 0) = 0.82). In general, as shown in Figure 5, they selected segments less often if the text remained on screen during question answering (inferential results: Est. = −0.99, CrI = [−2.18, 0.12], P(Est. < 0) = 0.96). However, when compared to presupposition, adults selected segments more often for implicature (inferential results: Est. = 1.65, CrI = [−0.33, 3.84], P(Est. > 0) = 0.95) when the text remained on screen during question answering (see Figure 5). Inferential results do not support this kind of interaction for the presupposition-irony comparison.

*3.3. Selection of Word Segments—Target Sentence*

Figures 6 and 7, respectively, show the proportions of selected segments depending on the different conditions for children and adults. Again, we only kept trials that were correctly answered and participants whose mean accuracy was higher than 50%. The color bars correspond to the three conditions: green corresponds to the texts with the presupposition condition, orange corresponds to the texts with the implicature condition, and violet to the texts with the irony condition. The panels in pink on Figure 6 correspond to each class and the panels in pink on Figure 7 correspond to whether the text remained on screen during question answering (left panel) or if there was only the question (right panel). For each figure, 1 means that participants selected at least one word segment from the context, and 0 means that participants did not select any word segment from the context. For example, in Figure 6, children from the first group (mean age: 6.6 y.o.) had a mean

selection of 0.57 in the ironic condition, meaning that they did not often select any segment compared to the presupposition condition, where their mean selection was 0.80.

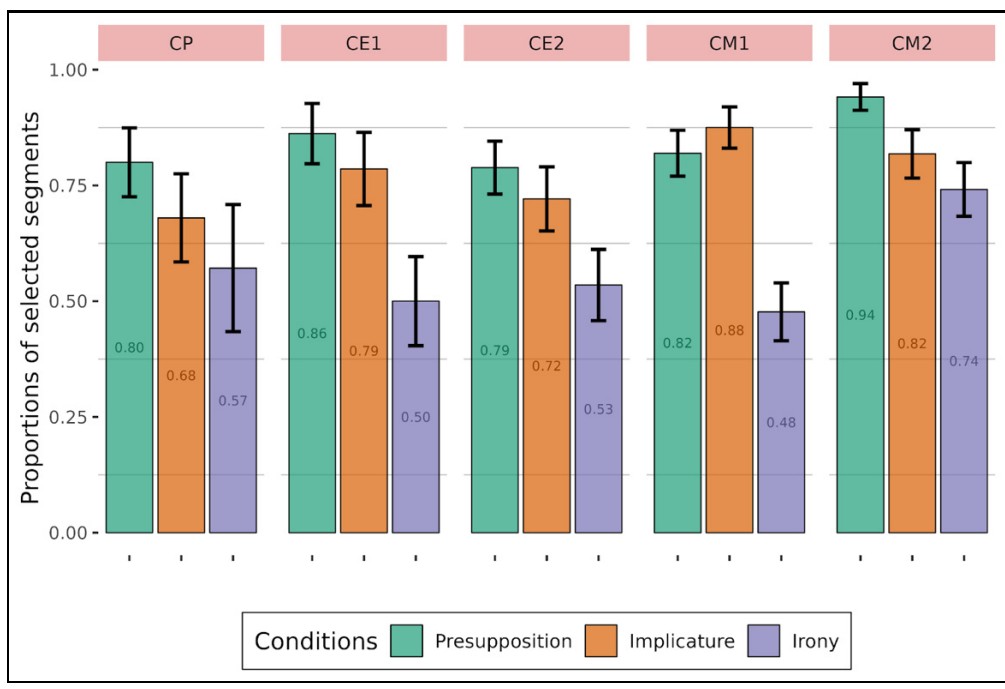

**Figure 6.** Proportions of segment selection depending on the type of implicitness and the class (children).

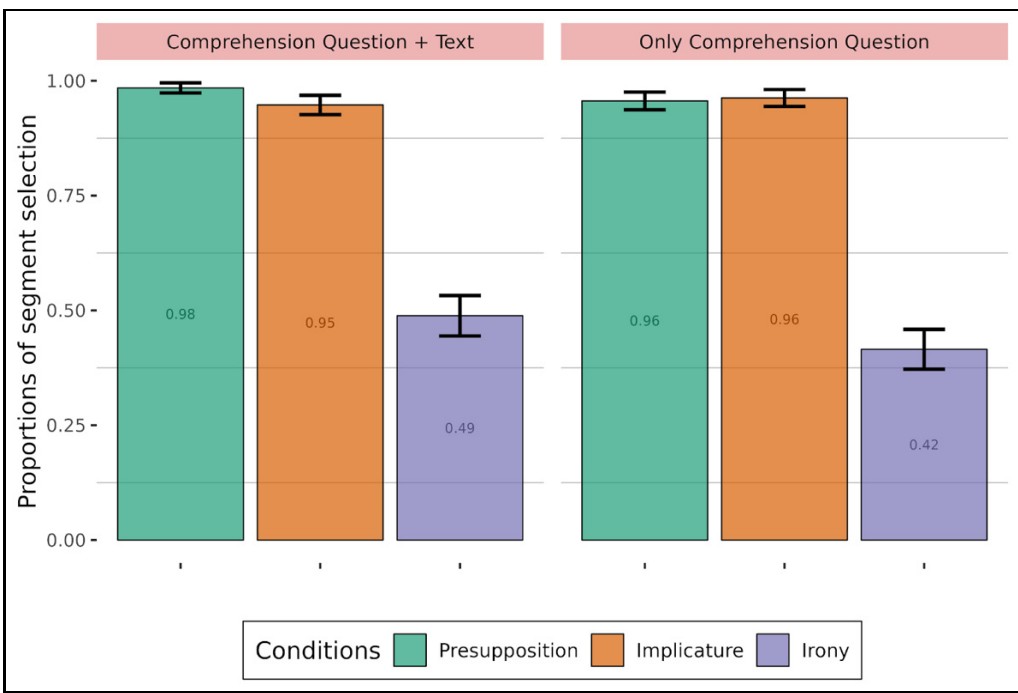

**Figure 7.** Proportions of segment selection depending on the type of implicitness and the presence or absence of the display of the text (adults).

As illustrated in Figure 6, in the target sentence, in general, children selected segments more often when they were in higher classes (the proportions of selection are getting higher in higher classes—the right panels—in Figure 6; inferential results: Est. = 0.35,

CrI = [−0.18, 0.91], P(Est. > 0) = 0.90). Compared to presupposition, children in general selected segments less often for irony (mean selection across class levels: 0.57 for irony and 0.85 for presupposition; inferential results: Est. = −2.55, CrI = [−4.11, −1.10], P(Est. < 0) = 1) and for implicature (mean selection across class levels: 0.79 for implicature and 0.85 for presupposition; inferential results: Est. = −0.74, CrI = [−2.42,0.91], P(Est. < 0) = 0.83). We did not find any relevant interaction with the class level.

Figure 7 shows that the adults selected segments less often for irony compared to presupposition in general (overall mean selection: 0.45 for irony and 0.97 for presupposition), which was confirmed by our statistical analysis (Est. = −6.55, CrI = [−10.23, −3.80], P(Est. < 0) = 1), but this was not the case for implicature compared to presupposition (overall mean selection: 0.96 for implicature and 0.97 for presupposition). Another finding was an interaction between the presupposition-implicature comparison and the presence or absence of display of the text during question answering (inferential results: Est. = −2.27, CrI = [−7.40,2.56], P(Est. < 0) = 0.83), meaning that the selection of segments for implicature compared to presupposition differed depending on whether the text was displayed or not during question answering (Figure 7). No relevant interaction between the presupposition-irony comparison and the text variable was found, and there was no relevant effect of the text variable in general.

### 3.4. Selection of Expected Word Segments for Presupposition and Implicature (Descriptive)

Figures 8 and 9, respectively, present the proportions of the correct selection of the expected word segments by children and adults. "Expected word", in the case of presuppositions, refers to the presupposition trigger (a lexical element in our examples, such as *continue* in example (1) above), and in the case of implicatures, we refer to expressions which evoke alternatives, generating a set of other statements which could have been preferred to the one which was pronounced. This is the case, in example (2) above, of the word *deux* (two), which is involved in a Horn scale (De Carvalho et al. 2016) and naturally evokes the stronger alternative *three*, hence the implicature "not three" and more generally "two and not more".

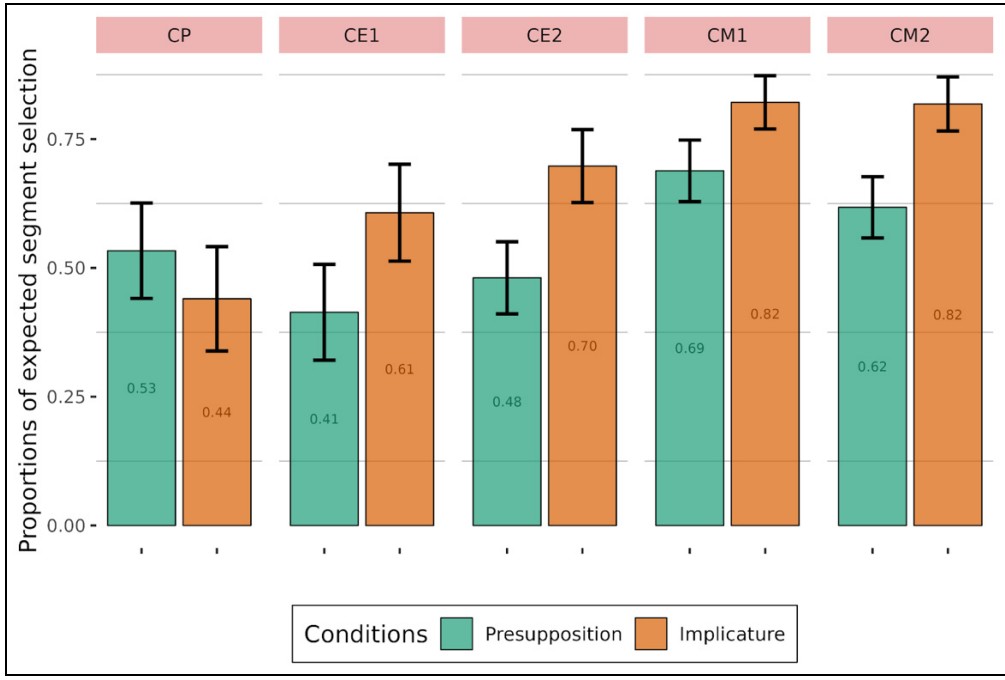

**Figure 8.** Proportions of expected segment selection depending on presupposition, implicature, and the class (children).

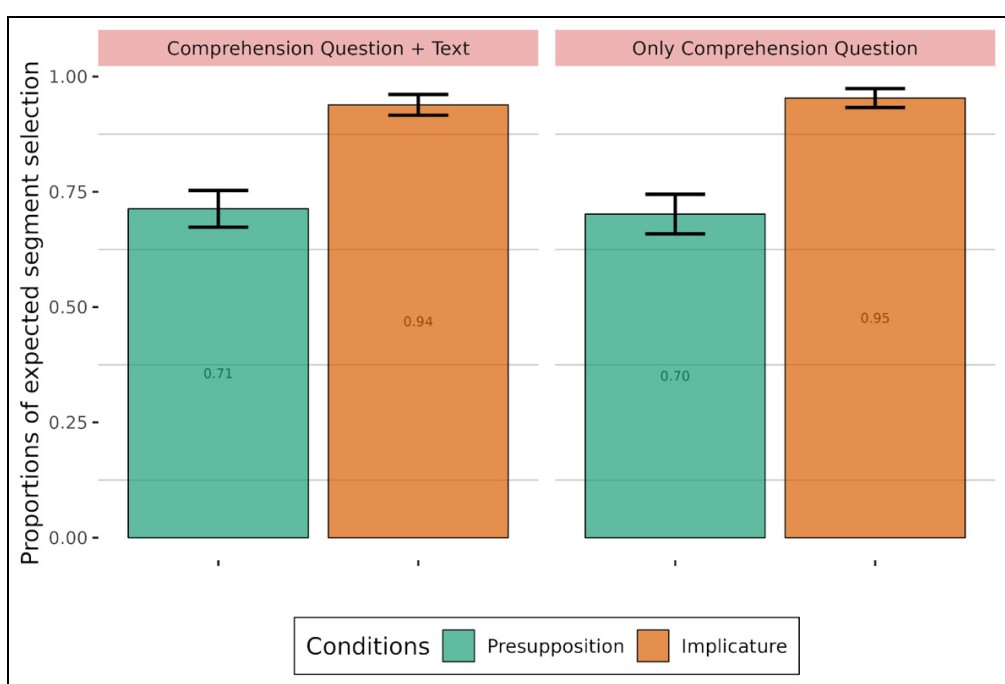

**Figure 9.** Proportions of expected segment selection depending on presupposition, implicature, and the presence or absence of the display of the text (adults).

We only kept trials that were correctly answered and participants whose mean accuracy was higher than 50%.

The color bars correspond to the two conditions: green corresponds to the texts with the presupposition condition, and orange corresponds to the texts with the implicature condition. The panels in pink on Figure 8 correspond to each class and the panels in pink on Figure 9 correspond to whether the text remained on screen during question answering (left panel) or if only the question was displayed (right panel). For each figure, 1 means that participants selected at least one word segment, and 0 means that participants did not select the expected segment. For example, in Figure 8, children from the first group (mean age: 6.6 y.o.) had a mean selection of 0.44 in the implicature condition, meaning that they did not often select the expected segment compared to the children in the last group (mean age: 10.7 y.o.), where the mean selection of the expected segment was 0.82 (so closer to 1).

As illustrated in the figures, adults (independent of the text variable) and older children seem to select the expected word segments more often for implicature than for presupposition.

## 4. Discussion

In this paper, we compared to what extent children and adults understand and interpret the three types of implicitness, i.e., presupposition, implicature, and irony.

For comprehension of implicitness, we found that, even though children at an earlier age have difficulty with all types of implicitness, especially irony (Figure 2), as they get older (e.g., at the end of primary school), they quickly understand them and the pattern that we found in children in higher classes resembled the one we found in adults (Figure 3). Interestingly, two observations emerge from Figure 3. First, the presupposition-irony and presupposition-implicature comparisons showed a difference depending on whether the text was displayed or not, with better accuracy when the text was not displayed for implicature and irony compared to presupposition. This difference is due to the fact that adults understand presupposition better when the text is displayed (0.86) than when it is not displayed (0.76). It thus seems that the text variable mainly has an influence on presupposition. A possible explanation is that the presupposed content is background content, which does not contribute to the main point of the utterance, the at-issue content

(cf. Tonhauser et al. 2018). This is not the case for the other two types of implicitness, which are part of the main topic of the discussion. This is particularly evident with irony since not understanding the implicit content in ironic statements results in misinterpretation.

The second observation is that implicature was less well understood among adults whereas irony seemed to be quite well understood (compared to presupposition), making implicature the most difficult type of implicitness in this experiment. In this respect, it should be pointed out that having the text on screen during question answering makes the interpretation of implicature worse for adults (again compared to presupposition). A possible explanation could be that when adults have the full text in front of them, they are more likely to stick to the (visible) literal meaning and hence more likely to refrain from drawing the pragmatic and defeasible inferences. More generally, during the comprehension task, participants may tend not to over-interpret the texts. Since conversational implicatures are non-monotonic inferences (they are only probable and can be canceled), participants may prefer to dutifully avoid them in their answers.

As for the selection of word segments, either in the context or in the target sentence (Figures 4–7), we noticed that, again, children behave in a similar way to adults when they get older, in this case at the end of primary school. However, when looking at the figures qualitatively (Figure 4 for children and Figure 5 for adults), children selected segments more often in general than adults.

Looking back at our hypotheses, presupposition should be detected on a local scale, meaning that participants should select segments especially within the target sentence and not outside. The results confirmed this for adults (Figures 5 and 7), and children at an older age (Figures 4 and 6). Indeed, in the target sentence, compared to presupposition, participants selected segments less often in the irony condition and the implicature condition (for children only). However, it was the other way around for the selection in the context, with participants selecting segments more often in the irony and implicature conditions compared to presupposition, both in adults and children. Phrases seem to be sufficient to interpret and understand presupposition, and hence a full comprehension of the target sentence does not appear as a necessary condition.

Regarding irony, it should be more difficult to detect it at a very local scale since the reader needs to realize a certain discordance between what is said and what is meant. The results showed that this seems to be the case: adults as well as children selected segments less often for irony compared to the other types of implicit in the target sentence (Figure 6 for children and Figure 7 for adults), while they selected segments more often in the context (Figure 4 for children and Figure 5 for adults). This is in line with the theory that irony cannot be detected and understood within a single statement only (i.e., one single sentence).

As for conversational implicatures, the hypothesis was that they should be more context-sensitive than presupposition, which is shown by the results found in the context: adults (especially when the text is displayed) and children did select segments more often in the implicature condition than in the presupposition condition, meaning that conversational implicature may be detected and understood on a less local scale than presuppositions in general.

Comparing the segments selected in the context with the segments selected in the target sentence enabled us to determine whether the understanding of the target sentence was sufficient for understanding each type of implicit. The results showed a difference between presupposition and implicature on the one hand and irony on the other. But to determine whether understanding the target sentence was necessary for understanding the implicit associated with presuppositions and implicatures, we checked whether the trigger for these types of implicit, what we called "the expected word segment", was among the segments selected by the participants. When looking descriptively at the presuppositions and implicatures used in the experiment (Figures 8 and 9), we observed that both adults and children (except for the first grade—CP) selected the expected word segments less often for presupposition than for implicature. Generally speaking, then, it appears that

both adults and children accurately detected presupposition, but they were likely to "miss" its actual linguistic trigger.

To understand these results, which our hypotheses did not allow us to anticipate, we looked at the scores item by item, for presuppositions and implicatures. Figures 10 and 11 show the results for presuppositions, respectively for all children (independent of the class variable) and adults. Figures 12 and 13 show the results for implicatures, respectively for all children (independent of the class variable) and adults. The color bars correspond to the types of presupposition (Figures 10 and 11) or implicature (Figures 12 and 13). For each figure, 1 means that participants selected at least one word expected segment, and 0 means that participants did not select the expected segment. For example, in Figure 10, children had a mean selection of 0.90 for the expected segment *fini* (finished), meaning that they more often selected the expected segment compared to the expected segments *ne plus* (no more) where the mean selection of the expected segments is 0.08 (closer to 0).

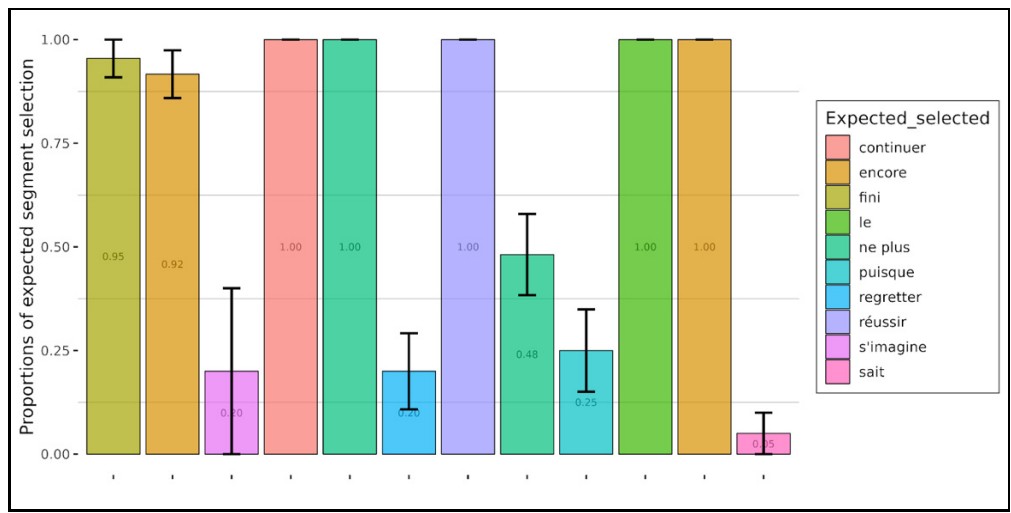

**Figure 10.** Proportions of expected segment selection depending on types of presupposition (children).

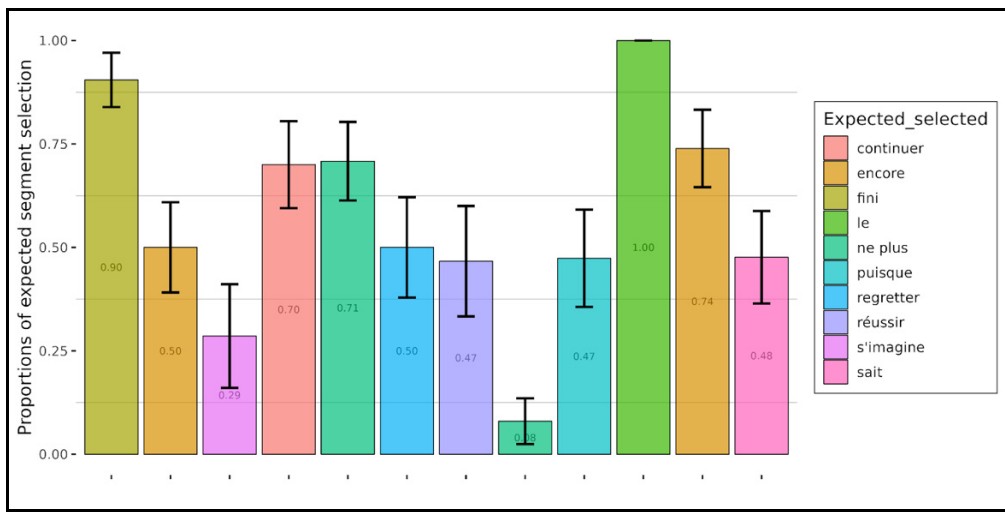

**Figure 11.** Proportions of expected segment selection depending on types of presupposition (adults).

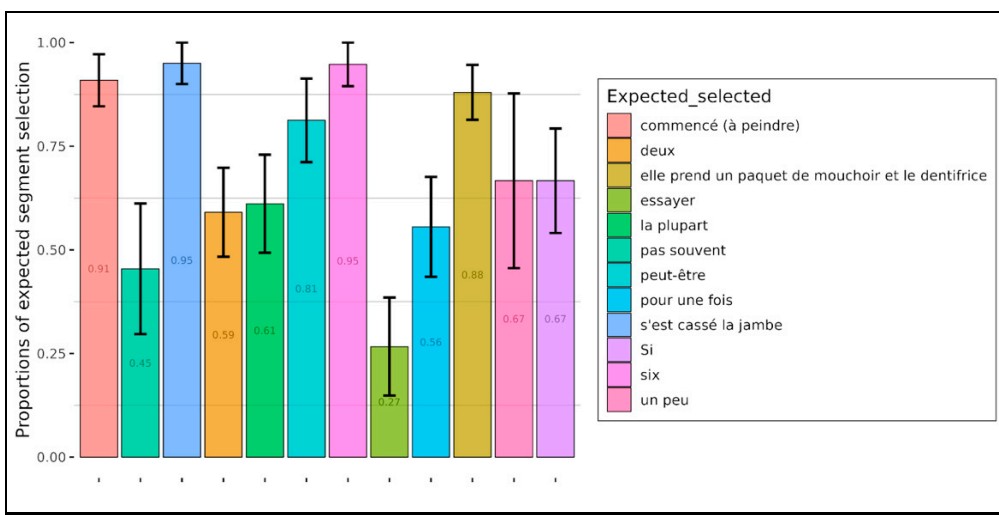

**Figure 12.** Proportions of expected segment selection depending on types of implicature (children).

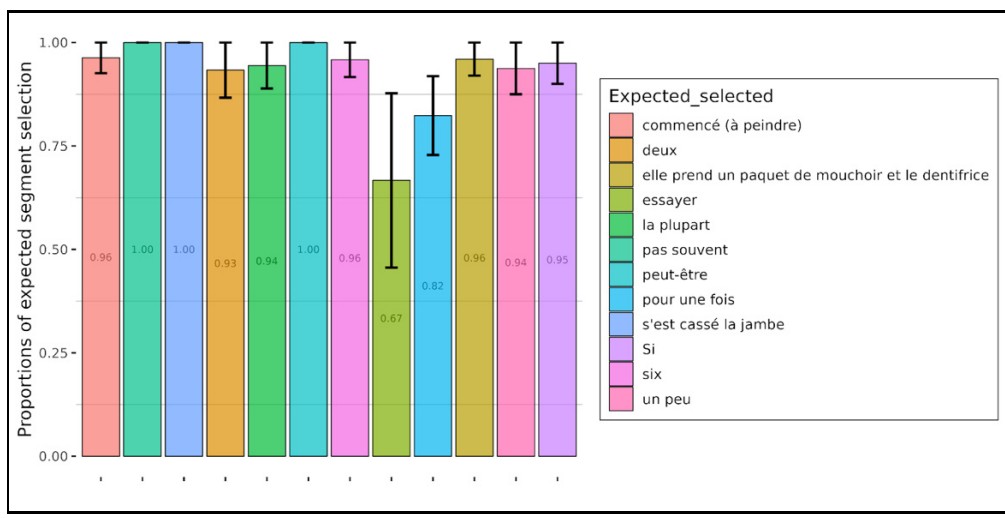

**Figure 13.** Proportions of expected segment selection depending on types of implicature (adults).

For presuppositions, a wide variation was observed depending on the triggers, and interestingly, the differences identified with children widened with adults, resulting in a bipartition of items among adults: on the one hand, items recognized as presupposition triggers (proportions close to 1) and on the other hand, items not recognized as such (proportions below 0.5, which can reach 0.05). This variation can be explained if we consider the following. First, it is not surprising that the participants selected other word segments than the trigger alone in order to justify their understanding; usually, the full phrasing of the propositional content that constitutes the presupposition includes large parts of the target sentence. For instance, the sentence *Tom continue à faire des progrès en calcul* ("Tom continues to progress in arithmetic") and the associated presupposition *Tom avait déjà fait des progrès en calcul* ("Tom had already made progress in arithmetic") share the material *Tom* and *faire des progrès en calcul*; these elements can naturally be perceived as clues that help to determine the presupposed content. Furthermore, the situation is even more striking when the trigger is just a factive operator: in such a case the presupposition merely amounts to the truth of a proposition already overtly given in the target sentence (*elle ne sait pas que Ludo déteste la soupe* 'She doesn't know that Ludo hates soup' presupposes the content of the subordinate clause *Ludo déteste la soupe* 'Ludo hates soup'). Accordingly, some participants may view the real trigger as playing a less significant role in the computation of the presupposition than the subordinate clause itself. In fact,

among the poorly recognized items were the factive verbs *savoir* 'to know' and *regretter* 'to regret', and the factive conjunction *puisque* 'since'. The counterfactual verb *s'imaginer* 'wrongly imagine', which we chose because it has been analyzed in detail by Ducrot (1968), was also poorly recognized, but it is easily confused with the verb *imaginer* 'imagine', which is not presuppositional, and its interpretation requires a very good knowledge of the French lexicon. Lastly, one of the two items including the negation *ne plus* 'not anymore' was not well recognized, but in this item, "ne plus" was in the scope of a modal verb "he shouldn't eat any more", which may have contributed to making it more difficult to calculate the presupposition: "he has already eaten some". In light of the above, it appears that the concept of presupposition, as characterized in theoretical linguistics, seems to cover two classes of triggers not equally well recognized by the participants: The first class includes factual and counterfactual triggers, often misidentified as triggers, and the other class includes, among others, aspectual triggers. On this topic, readers are referred to the literature on strong and weak presupposition triggers (see Cummins et al. (2013); Romoli (2015)).

As far as implicatures are concerned, we can see that their triggers are very unevenly recognized by children, while adults recognize them well. There are just two exceptions, one of which is the verb *essayer* 'to try'. But in the item, this verb was in the present tense and participants were asked whether the target sentence generated the implicature "won't succeed". Even if this implicature exists, it is quite easily canceled out, which is probably why it was less frequently recognized than the other implicatures in the experiment. It would have probably been different with a verb in the past tense. The other exception is for the expression *pour une fois* 'for once', and the expected implicature 'not being used to'. This implicature was less well recognized than the others, but it was still fairly well recognized (above 0.8). This lower score is perhaps due to the fact that 'for once' in the target sentence and 'not being used to' in the comprehension question are two expressions that do not belong to the same syntactic paradigm and cannot be placed, as such, on a Horn scale. This could explain why participants had more difficulty generating an implicature with this item.

## 5. Conclusions

Our main conclusion is that our initial hypotheses seem to be confirmed by the results of our experiment. The results show a clear local anchoring for presuppositional implicitness and a strong appeal to the context to interpret irony. This tends to show that the concept of a sentence, as a "semantic unit", is poorly relevant and operational when it comes to understanding various kinds of implicit meaning. Indeed, the full comprehension of the target sentence never appears as a necessary condition to understand presuppositions; on the other hand, it never appears as a sufficient condition to recognize irony. This is even more noticeable with the understanding of conversational implicatures where the full comprehension of the sentence appears as neither necessary (implicatures are mainly attached to precise lexical material) nor sufficient (contextual information helps to consolidate the defeasible inference).

The comparison between children and adults also shows that by 5th grade, pupils are progressing to a level of comprehension comparable to that of adults, on the one hand, and that their most significant progress concerns implicatures and irony, on the other hand. This observation suggests that the teaching of comprehension at school would benefit from shifting the emphasis from working on the sentence to focusing pupils' attention on the role of context in the shaping of meaning.

Moreover, the experiment pointed out that while presuppositions are fairly well recognized by both adults and 5th grade pupils, their triggers are not detected. This failure in detection highlights the importance of distinguishing between vocabulary breadth (how many words pupils know) and depth of vocabulary (what pupils know about those words) (Ouellette 2006; Tannenbaum et al. 2006) in order to improve at school the understanding of the lexical semantics of presuppositional triggers that are poorly recognized as such by

pupils. Our study also showed that more precise experiments on presuppositions could be carried out in order to refine analyses of so-called weak or strong presupposition triggers. This finer-grained analysis would then make it possible to better explain the detection failures observed for verbs such as "know" or "regret".

**Author Contributions:** The four authors C.P., C.B., L.R. and B.G.-W. designed the study and wrote the paper together. C.P. programmed the experiment and analyzed the data. B.G.-W. ran the experiments with children while C.P. ran the experiment with adults. All authors have read and agreed to the published version of the manuscript.

**Funding:** This project received financial support from the CNRS through the MITI interdisciplinary programs.

**Institutional Review Board Statement:** This study was conducted in accordance with the Declaration of Helsinki and approved by the Institutional Review Board (or Ethics Committee) of CNRS (protocol code 2-23014 and date of approval: 24 January 2023) for studies involving humans.

**Informed Consent Statement:** Informed consent was obtained from all subjects involved in this study.

**Data Availability Statement:** Data supporting reported results can be found in HUMANUM. All the code and analyses are available at https://osf.io/4gh6c/ (accessed on 12 December 2023).

**Conflicts of Interest:** The authors declare no conflicts of interest.

## Notes

1    This has been extensively addressed in the pragmatic literature from various perspectives; see for instance Ducrot (1972), Blakemore (1987), Wilson and Sperber (1993), and Recanati (2003).

2    The literature on implicature, and particularly on conversational implicatures and scales, is enormous and still growing. We cannot summarize it here, but interested readers can refer to the two encyclopedia articles by Simons (2012) and Davis (2019). They will provide an overview of the wealth and topicality of research on implicatures.

3    It should be noted however that there also exist cases of irony where the speaker does not overtly utter what she believes to be false (see, for instance, the so-called irony without flouting and non-declarative irony, in Garmendia 2018). In our paper, we only looked at irony as saying something and meaning the opposite. However, the general definition of irony is more complex and is not restricted to this one.

4    A context will be considered favorable if certain elements of the situation are sufficiently salient for the child to pay attention to them and infer the presence of an implicit (as in the case of "How clean you are!" in a situation where the child perceives that he is very dirty) or if the statement containing an implicit is part of a routine well integrated by the child (if brushing teeth always precedes going to bed, the child is able to respond with the conversational implicature "I'm not sleepy" when he hears "Go and brush your teeth").

5    At this school, 55% of the pupils come from advantaged backgrounds, 30% from average backgrounds, 10% from modest backgrounds, and 5% from disadvantaged backgrounds.

6    This means that we did not have the same number of children per list.

7    Because of the difference found in accuracy between conditions, the number of observations between conditions is not well counterbalanced. We did regression models with a random structure to take into account this variability.

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
