# Peer review of "How Relevant Is the Sentence Unit to Accessing Implicit Meaning?"

_languages, doi:10.3390/languages9020042_

Round 1

Reviewer 1 Report

Comments and Suggestions for Authors

The article presents an interesting and useful theme for teaching reading. It is well researched and presented, and the experimental research is well conducted and explained. The results are certainly interesting, but expected given the very definition of the three forms of implicit investigated. 

However, we might question the formulation of the hypotheses which, in our opinion, should have emphasised the link between understanding the proposed sequences rather than the a priori link between the meaning of these sequences and the form of implicit concerned. 

We suggest that in formulating the hypotheses and confirming the results obtained, the author should insist on this link between the comprehension - interpretation of meaning and the implicit involved.

Author Response

We sincerely thank the reviewers for their helpful and thought-provoking comments. We have found the questions and criticisms posed very constructive, and believe that incorporating them has led to an improved version of this revised manuscript. We have done our best to incorporate our answers to reviewers into a revised manuscript in a way that is compatible with both the page limit given to us and the theme of the issue in which the article must be inserted.

Our specific responses to each comment are given below (except for the typos and the figures numbering that we directly corrected in the paper); the original reviewer text is in gray.  As some of the remarks, without being absolutely identical, are shared by several reviewers, we have sometimes grouped into a sole response to several comments.  Rather than repeating this response several times, we will refer to its first occurrence.

Reviewer 1

The article presents an interesting and useful theme for teaching reading. It is well researched and presented, and the experimental research is well conducted and explained. The results are certainly interesting, but expected given the very definition of the three forms of implicit investigated. 

However, we might question the formulation of the hypotheses which, in our opinion, should have emphasised the link between understanding the proposed sequences rather than the a priori link between the meaning of these sequences and the form of implicit concerned. 

We suggest that in formulating the hypotheses and confirming the results obtained, the author should insist on this link between the comprehension - interpretation of meaning and the implicit involved.

We described the expected comprehension for each type of implicit more in the operational hypotheses (=predictions on page 5). Still, we changed the theoretical hypotheses section, and we hoped it will be clearer this way now.

Reviewer 2

- The author(s) assume much more familiarity with the French educational system than what should be taken for granted. I would expect a (say: South American) reader with average motivation to read the first page, be confused, and give up. The author(s) should be reminded that this article is in English, and for an international audience, and therefore, it should be assumed by default that the reader has *no* particular idea of what the French school system looks like.
- In its current form, is not clear to the reader who reads the article for the first time how the beginning of the introduction is relevant for the theoretical hypotheses the article presents (once one arrives at p. 5, one gets a hunch, and it becomes clearer at the end, but this should be made more explicit). However, I am generally not convinced that the study of the comprehension of the implicit has much to say about how grammar should be dealt with (or not) in elementary school.

We have completely rewritten this part of the text.

  Maybe that there is a point to be made, but it has to be done much more explicitely: There is a cool study with interesting data, and there is a societally important question, but how the two go together is not obvious. I would suggest to focus on the data, and leave aside the question of grammar teaching in school.

We followed the recommendation by putting aside the discussion about grammar teaching.

- The authors discuss presuppositions as purely lexically triggered, and conversational implicatures as not being lexically triggered. There may be something to it, but I believe it would be better to separate on one hand
  + types of inferences (presupposition vs. implicatures etc.)
  + triggers of inferences (lexical triggers, general conversational principles, etc.)
  One issue is that what is called /hard/ presupposition triggers have an *obligatory* anaphoric dependence on the context, which does not square easily with the assumptions developped here, and *soft* presupposition triggers also *may* have an anaphoric dependence on the context. So, even if the trigger is local, the antecedent may be more important, and not be local.    

Thanks for  that suggestion, it is particularly relevant to our work. In a way this idea was part of what we were dealing with but we didn't make it visible enough. We tried to bring more clarification about it in the text. Also it should be noted that the experiment was designed so that no presupposition had an overt antecedent in the context (the presuppositions had to be globally accommodated). For instance in the sequence with “Tom continue à faire des progrès en calcul” there is, in the context, no mention of the fact that Tom had already made some progress”. The reason was to ensure that these presuppositions appear as *implicit* contents. So any obligatory anaphoric dependence on the context was intentionally avoided in the experiment, in order to only focus on the participants’ ability to retrieve the implicit content from purely linguistic clues. We added a precision about it in S2.2. 

- Generally, I find it sometimes difficult to understand what Figures are supposed to show. These should be better explained.

We tried to explain in a better way what the figures correspond to by adding an introduction paragraph before the descriptive and statistical analyses.

- It is not very useful for readers not familiar with the French school system to read about CP - CE2, and cycle 2 and cycle 3. This either needs to be explained better, or be translated differently.

We have made the necessary changes by indicating the age of the pupils and the correspondence between the different French levels and grades.

- p. 4 (l. 182ff.): I found the presentation of how the hypotheses are tested confusing. Maybe take one or two examples, and walk the reader through?
See response to Reviewer 1.

- p. 5, l. 236: what does *counterbalanced* mean here? Explain!
We added a clarification sentence about that.

- p. 5: what kind of school was this? Public? Private? Which arrondissement or area? Data on socioeconomic status of parents (could be gathered from what children pay for the canteen)? 

We have added all the information requested in the text and in the notes.

- p. 5, Table 1: column title should not be classes only, but also ages (CP=6 years); you want this to be comprehensible immediately for people outside of France.

We changed that (and we also changed the graphs for this point).

- p. 6: I find the case of irony not that obvious: It is a rhetorical question, contains a presupposition trigger ("toujours"), and it relies much on pragmatic reasoning that could probably be eliminated (e.g., "Tom's father says to Tom's mother: I see that Tom is top of the class", or something the like)
It is not clear why the presupposition trigger and the rhetorical question pose a problem here. Ironic contents can be located in presuppositions (e.g. “I realize that you’re a very nice person” said to someone being mean). Irony can occur in non-declarative statements (see Wilson 2006:Lingua 116, Garmendia 2018) and one can argue that the fact that (4a) can be perceived as a rhetorical question (although not a classical one) is due to the irony it contains (and not the other way round). However we added some clarification to indicate that we only focused on that kind of irony which must obligatorily be recognized in order not to yield an incoherent discourse. 

- The segmentation of sentences presented to subjects strikes me sometimes strange. For instance, why is "de campagne" one word, rather than two? And if we go for higher units (which is legimitate), why is "maison de campagne" not one unit?

When we designed the experiment, our hypotheses focused on whether the participants selected segments from the context sentence or the target sentence depending on the type of implicit. Also, because the items were rather long, having a strict word by word segmentation would have been fastidious for participants. Moreover, for some conditions, the trigger segment was sometimes on words such as a determiner (“le” for presupposition for example). So we decided to segment some sentences the following way: i) we had fine-grained segmentations for some items depending on the conditions (whether there was a small trigger word) ii) we did not do word by word segmentation all the time so that the drag-and-drop task would not be too difficult, especially for children.

- Could the drag-and-drop requirement have caused subjects to choose shorter and incomplete fragments as answers?

According to the teachers, the drag-and-drop task was a good task for the children, which is why we decided to go with it. It is possible that this requirement may have caused participants to choose shorter and incomplete segments, but still, we were only interested in i) which part of the text the participants selected segments; ii) whether they selected the expected segment for presupposition and implicature. 

- Among those who answered the question correctly, what proportion did use actual sentences to answer the question, or something that might be considered a sentence (i.e., something having identifiable truth conditions)?
  When I look at the raw data, it looks like (at least) the children didn't select sentences, but only extracts of sentences for their justification: "Alexandre s'est acheté grande maison de campagne" if I interpret correctly the first line of the csv file. While not a grammatical sentence, this still is truth-evaluable (and so, a kinda-sentence). Is this a general pattern in the data, and can something  be said about this (because this could give us some insight on what subjects see as a minimal unit of information)

We agree and accordingly we didn’t pay much attention to the cases where the participants would select full sentences or almost complete sentences because we expected that these cases wouldn’t be really significant (given that, in addition, the drag-and-drop procedure might be tedious for the participants). This was justified in §1 by “Note that to diagnose the necessary nature of sentence comprehension, we did not expect participants to select every segment in the sentence. We left the participants free to consider that certain segments might be of secondary importance in justifying their responses”.

- Among those who answered the question correctly, but did not choose the expected segment, what did they choose? In other words, is there any sense to be made of these answers, maybe that they didn't choose the trigger rather than the propositional content, or is this a sign that they were lucky with their 50% chance of hitting a true answer?
  It might be that this simply shows that people do not take the trigger to be as relevant as some other information (e.g., for "John knows that Martha stole the rabbit", the authors would want the subjects to identify "KNOWS", but maybe subjects believe that the real relevant information is "MARTHA STOLE THE RABBIT" [which is the content of the presupposition, rather than the trigger]) - seeing that identification of "SAIT" as trigger is absolutely abysmal at least in adults (Figure 10)
We agree with this remark. But, if we understand it correctly, this very point was already addressed in §4 when we discuss the results on presuppositions and especially the case of factive triggers (such as ‘know”, ‘regret”, ‘since”).

Such an analysis should be easy to do given the collected data, and if there is something consistent here, it might give us insight into the folk understanding of untrained experimental subjects. Even if there is nothing consistent to be found, it still might be relevant to eliminate speculation about different types of presupposition, for instance.

Thank you for this suggestion which is interesting, but which would take us away from our main objective and would require a change in methodology. Indeed, the experiment we ran was designed for quantitative and not qualitative analyses of collected data.

Reviewer 3

Page 1: you should say what CP corresponds to (first year of elementary school)

Dans le premier para de l’intro

See response to Reviewer 2.

Page 1: perhaps specify that these are the first three years of elementary school and the following three are the last two years and the first year of middle school

Dan le para “From cycle 2, devoted to…”

See response to Reviewer 2.

Page 2, L75: the procedural/conceptual meaning distinction within Relevance Theory may be relevant for this discussion, see for instance

Escandell-Vidal, Victoria, Manuel Leonetti & Aoife Ahern. 2011. Introduction: Procedural Meaning. In Current Research in the Semantics/Pragmatics Interface, vol. 25, xvii–xlv. Bingley: Emerald Group Publishing. https://doi.org/10.1108/S1472-7870(2011)0000025004.

Dans le paragraphe, pronoun or the quantification domain...

Thanks for this observation. The procedural/conceptual distinction is indeed related to what is at stake in the paper, and we added some references in order to make the connection more apparent. However we noted that the experiment was not oriented to shed a specific light on this division of tasks (for instance presuppositions can often be seen as pertaining to both the conceptual and the  procedural dimensions); so we chose not to develop any further, just to avoid any confusion or misleading discussion from our part.

Page 2, L85 : You want to show that even if the sentence is a semantic unit, its semantic content is retrieved in great part via the context, so the notion of sentence is not useful for pupils to understand a text. But the notion of sentence is not given to pupils to understand a text, but to understand how language works (acquire metalinguistic skills). Also, I don't see how your work contributes to the confusion described at the end of the preceding paragraph between sytnax and semantics.

Page 3, L109 : I find this explanation insufficient. These are scalar implicatures (you should say that, and provide some references of the immense literature on them) so they are based on a scale.

L111: perhaps you can say a bit more here too. These are *conversational* (and not conventional, for instance) implicatures, so, according to Grice, they are the result of a calculation, a reasoning based on the fact that speakers are supposed to obey the maxims of conversation and the cooperative principle.

We developed this part and tried to make it clearer.

Page 3, L130: I would not say that it looses it's compositional meaning at all. It also keeps its literal meaning, nothing in the sentence indicates that the sentence means something different. It is the *speaker* that, with that sentence, means something different *in that particular context*.

Page 3, L145: this is not the opposite of (4a), if by "opposite" you mean "opposite polarity". If not, what do you mean by opposite? Perhaps to rely on previous work on irony would be helpful

We agree with these 2 remarks, and we reformulated the part in a, hopefully, more accurate way.

Page 4, L162: according to Grice, irony too is the result of a conversational implicature (violation of the maxim of quality)

Indeed, subsequent works on irony have convincingly argued that Grice’s analysis of irony as conversational implicature is unsatisfactory and somehow faulty (a.o. implicatures usually just add complementary meanings but never bring about substitutions of meanings). The arguments are summarized in Garmendia (2018). That is why we chose to treat irony apart.

Page 6: not sure it is the correct translation (his father said to him: so, still top of the class)

The translation is correct.

I'm doubtful about the capacity of CP pupils to understand and perform this experiment.

According to the teachers, pupils should have been able to understand and perform the experiment. An experimenter explained everything to the children, helped them perform the task and was with them the whole time. Still, it is true that CP pupils had difficulty compared to the others (and the results showed that in a way). 

Reviewer 2 Report

Comments and Suggestions for Authors

This article contains an empirical study of 3 different types of implicit contents, namely presuppositions, conversational implicatures and irony. It is tested whether children can identify such implicit content, and also, what they believe helps identifying these implicit contents. The underlying issue the authors seem to address is whether the French school system's preoccupation with the sentence (caracterized formally) is wise, or whether this should be amended to a wider notion of context. I don't believe that the article makes a convincing point with respect to this issue. The study itself is interesting, and it will make for an interesting article, but I don't recommend it to be published in its current form.

# General Remarks

- The author(s) assume much more familiarity with the French educational system than what should be taken for granted. I would expect a (say: South American) reader with average motivation to read the first page, be confused, and give up. The author(s) should be reminded that this article is in English, and for an international audience, and therefore, it should be assumed by default that the reader has *no* particular idea of what the French school system looks like.
- In its current form, is not clear to the reader who reads the article for the first time how the beginning of the introduction is relevant for the theoretical hypotheses the article presents (once one arrives at p. 5, one gets a hunch, and it becomes clearer at the end, but this should be made more explicit). However, I am generally not convinced that the study of the comprehension of the implicit has much to say about how grammar should be dealt with (or not) in elementary school.

  Maybe that there is a point to be made, but it has to be done much more explicitely: There is a cool study with interesting data, and there is a societally important question, but how the two go together is not obvious. I would suggest to focus on the data, and leave aside the question of grammar teaching in school.
- The authors discuss presuppositions as purely lexically triggered, and conversational implicatures as not being lexically triggered. There may be something to it, but I believe it would be better to separate on one hand
  + types of inferences (presupposition vs. implicatures etc.)
  + triggers of inferences (lexical triggers, general conversational principles, etc.)
  One issue is that what is called /hard/ presupposition triggers have an *obligatory* anaphoric dependence on the context, which does not square easily with the assumptions developped here, and *soft* presupposition triggers also *may* have an anaphoric dependence on the context. So, even if the trigger is local, the antecedent may be more important, and not be local.    
- Generally, I find it sometimes difficult to understand what Figures are supposed to show. These should be better explained.

# More Specific Remarks

## On the Text

- It is not very useful for readers not familiar with the French school system to read about CP - CE2, and cycle 2 and cycle 3. This either needs to be explained better, or be translated differently.
- p. 3: "theyshe" - is this genderfair 2.0 or an error?
- p. 3: "the addresssee loos outside the limits ..." -> HAS TO LOOK?
- p. 4 (l. 182ff.): I found the presentation of how the hypotheses are tested confusing. Maybe take one or two examples, and walk the reader through?
- p. 5, l. 236: what does *counterbalanced* mean here? Explain!
- p. 5: what kind of school was this? Public? Private? Which arrondissement or area? Data on socioeconomic status of parents (could be gathered from what children pay for the canteen)?
- p. 5, Table 1: column title should not be classes only, but also ages (CP=6 years); you want this to be comprehensible immediately for people outside of France.
- p. 6: I find the case of irony not that obvious: It is a rhetorical question, contains a presupposition trigger ("toujours"), and it relies much on pragmatic reasoning that could probably be eliminated (e.g., "Tom's father says to Tom's mother: I see that Tom is top of the class", or something the like)

## On Data (and Analysis)  

- The segmentation of sentences presented to subjects strikes me sometimes strange. For instance, why is "de campagne" one word, rather than two? And if we go for higher units (which is legimitate), why is "maison de campagne" not one unit?
- Could the drag-and-drop requirement have caused subjects to choose shorter and incomplete fragments as answers?
- Among those who answered the question correctly, what proportion did use actual sentences to answer the question, or something that might be considered a sentence (i.e., something having identifiable truth conditions)?

  When I look at the raw data, it looks like (at least) the children didn't select sentences, but only extracts of sentences for their justification: "Alexandre s'est acheté grande maison de campagne" if I interpret correctly the first line of the csv file. While not a grammatical sentence, this still is truth-evaluable (and so, a kinda-sentence). Is this a general pattern in the data, and can something  be said about this (because this could give us some insight on what subjects see as a minimal unit of information)
- Among those who answered the question correctly, but did not choose the expected segment, what did they choose? In other words, is there any sense to be made of these answers, maybe that they didn't choose the trigger rather than the propositional content, or is this a sign that they were lucky with their 50% chance of hitting a true answer?

  It might be that this simply shows that people do not take the trigger to be as relevant as some other information (e.g., for "John knows that Martha stole the rabbit", the authors would want the subjects to identify "KNOWS", but maybe subjects believe that the real relevant information is "MARTHA STOLE THE RABBIT" [which is the content of the presupposition, rather than the trigger]) - seeing that identification of "SAIT" as trigger is absolutely abysmal at least in adults (Figure 10)

  Such an analysis should be easy to do given the collected data, and if there is something consistent here, it might give us insight into the folk understanding of untrained experimental subjects. Even if there is nothing consistent to be found, it still might be relevant to eliminate speculation about different types of presupposition, for instance.

Comments on the Quality of English Language

English is globally ok (with minor typos); problems with French inside perspective.

Author Response

(The authors gave the same response as above.)

Reviewer 3 Report

Comments and Suggestions for Authors

The experiment is interesting, but I miss the link with the main research question about the notion of "sentence" in grammar teaching in elementary school.

See more comments directly on the manuscript.

Comments on the Quality of English Language

I strongly invite to revise English grammar and wording, especially in the introductory part.

Author Response

(The authors gave the same response as above.)

Round 2

Reviewer 2 Report

Comments and Suggestions for Authors

My issues concerning the presentation of the French school system, accommodation vs. non-accommodation of presuppositions, presentation of figures, etc. have been addressed. So, I withdraw my reserves as to the publication.

Dear author(s): If you were given this text without the first and the last paragraph, would you feel it to be incomplete or incoherent in any way? Would you think any better or any less of the article?

You seem to really want to include it, but I feel it confuses the reader, and adds nothing that would be addressed in the main part of the article. To me, it feels like an avoidable red herring, unnecessarily bracketing an article that is only very vaguely related to the issue of teaching a formal definition of the sentence.

NB: I agree that "grammar teaching" in French schools is stupid, but this article may not be the place to state the obvious. I doubt that the relevant people in the ministry care enough to read an article in English, and even if they do, their judgment that children cannot be confused with grammar or linguistics has already been made.

Comments on the Quality of English Language

p. 2, l. 86-87: "the addressee look outside..." -> "_has to_ look outside"?

Author Response

Dear Expert,

Thank you for your constructive criticism. We have taken your advice and deleted the first and last paragraphs.

Thank you for reading this article so carefully.